# Genetic Comparisons of Body Weight, Average Daily Gain, and Breast Circumference between Slow-Growing Thai Native Chickens (Pradu Hang dum) Raised On-Site Farm and On-Station

**DOI:** 10.3390/vetsci10010011

**Published:** 2022-12-25

**Authors:** Nitiporn Chaikuad, Wipas Loengbudnark, Vibuntita Chankitisakul, Wuttigrai Boonkum

**Affiliations:** 1Department of Animal Science, Faculty of Agriculture, Khon Kaen University, Khon Kaen 40002, Thailand; 2Network Center for Animal Breeding and Omics Research, Faculty of Agriculture, Khon Kaen University, Khon Kaen 40002, Thailand

**Keywords:** growth performance, genetic parameters, indigenous chicken, estimated breeding value, selection index

## Abstract

**Simple Summary:**

Native chickens are vitally important livestock at the community level and represent food security and a source of savings, especially in developing countries where they are considered a valuable genetic resource for use in the development of high-yielding breeds. However, due to the constraints of their slow growth rate, there is an inadequate supply to meet consumer demand. Therefore, genetic improvement for growth traits is one sustainable way to solve these problems. In this study, a multiple trait animal model and selection index are developed as a solution for this problem to improve growth performance in smallholder farms.

**Abstract:**

To ensure that any new technology developed within an experimental station is appropriate to the community’s needs and compatible with the existing systems, on-site farm research is an important component in examining the effectiveness of agricultural research. The present study examined the growth performance and genetics of Thai native chickens under conditions typically experienced by farmers on smallholder farms (on-site farms) compared with at an experimental unit (on-station). There were 1694 Thai native chickens (Pradu Hang dum) used in this experiment, and they were divided into 613 chickens for the on-station and 1081 chickens for the on-site farm experiments. The individual chicken data included the birth weight (BW0) and body weight at 4, 8, 12, and 16 weeks of age (BW4, BW8, BW12, and BW16, respectively), ADG from 0–4, 4–8, 8–12, 12–16 weeks of age (ADG0–4, ADG4–8, ADG8–12, ADG12–16, respectively), and breast circumference at 8, 12, and 16 weeks of age (BrC8, BrC12, BrC16, respectively). A multiple traits animal model and a selection index were used to estimate the variance components, genetic parameters, and breeding values of growth traits. The results showed that the body weight, average daily gain, and breast circumference at 8, 12, and 16 weeks of age of Thai native chickens raised on-station were higher than those raised on-site at the farm among mixed-sex and sex-segregated chickens, while the birth weight and body weight at four weeks of age (BW0 and BW4) and ADG from 0–4 weeks of age (ADG0–4) were not significantly different (*p* > 0.05). The heritability estimates of body weight, average daily gain, and breast circumference in the chickens raised at the on-site farm and on-station were moderate to high, with on-station values slightly higher than on-site farm values. The heritability estimates of body weight were 0.236 to 0.499 for the on-site farm, and 0.291 to 0.499 for on-station. For average daily gain, the heritability estimates were 0.274 to 0.283 for the on-site farm and 0.298 to 0.313 for on-station; meanwhile, and for breast circumference, the heritability estimates were 0.204 to 0.268 for the on-site farm and 0.278 to 0.296 for on-station. Both phenotypic and genetic correlations among and between growth traits were positive and ranged from low to high values. The top 20% of the estimated breeding values and selection indices in the on-site farm and on-station experiments showed that the body weight at eight weeks of age (BW8), ADG from 4–8 weeks of age (ADG4–8), and breast circumference at eight weeks of age (BrC8) should be used as selection criteria for Thai native chicken breeding programs. In conclusion, the genetic parameters and breeding values in on-station experiments showed that the breeding program by selection index for improving growth performance is valid. Therefore, to implement such a breeding program in an on-site farm, an intensive or semi-intensive farm system should be considered to minimize the effect of genotype-environment interaction.

## 1. Introduction

Native chickens are vitally important livestock at the community level and represent food security and a source of savings, especially in developing countries [1,2,3,4]. The main advantages of native chickens are that they are easy to raise and well adapted to local conditions, especially to the use of local feed source, and have an excellent ability to resist disease and poor weather compared to exotic chicken breeds [5,6]. From a marketing perspective, native chicken meat has proven itself in many ways, such as its distinct taste and flavor [7], and is used as a health food because of its low cholesterol [8] and high nutritional components [9,10,11], which are essential human health considerations. Therefore, native chicken meat has created a rapidly increasing demand in the chicken market for consumers [12].

Although the benefits of native chickens are distinctive in many respects, they present a significant limitation that affects their production worldwide: their growth rate is slow compared to that of commercial broilers and crossbred native chickens. The slower growth rate significantly affects feed conversion as well. At the same time, most Thai indigenous chickens are raised in backyards with poor husbandry practices, inadequate nutrition, and hot environments. With this system, the farmers must spend a long time raising chickens to achieve the weight requirements of the market [13,14]. Previous studies by Tongsiri et al. [14] revealed that it takes 14–16 weeks for Thai native chickens to reach an average body weight of 1.2–1.5 kg, which is the market weight. Moreover, Korean native chickens take ten weeks to reach approximately 2.0 kg [15] while East African chickens take 19–21 weeks [1]. The different management in each area [16,17] and shortage of good genetic animals for the next generation is another major obstacle for farmers

To solve this problem, a genetic method for improving growth performance should be one of the most effective and sustainable methods. Therefore, the Thailand Research Fund (TRF) and the Thai Department of Livestock Development (DLD) have cooperated in collecting a population of Thai native chicken breeds for conservation and future utilization since 2001. The Pradu Hang dum chicken breed was designated in the project and is now raised throughout Thailand. In 2010, the Research and Development Network Center on Animal Breeding (NCAB), Faculty of Agriculture, Khon Kaen University, was established to conduct research on genetic improvements of Thai native chicken breeds, including growth and carcass characteristics [18,19,20], egg production [21], and fertility traits [22], based on knowledge of both quantitative genetics and molecular genetics. Consequently, the Thai native chicken has been genetically improved after seven generations, and the breed is called “Pradu Hang dum KKU55”.

To ensure that “Pradu Hang dum KKU55” which has been genetically improved in its growth performance after seven generations within our experimental station is appropriate to the community’s needs, and compatible with existing systems, the on-site farm (smallholder farmers) is an important component in examining the effectiveness of breeding programs. In the past, several studies have compared the effects of farming systems (intensive and extensive) on genetics parameters [23,24,25,26]; however, study of the genetics of each individual chicken which is the ultimate goal for genetic selection, has not been investigated. It seems possible that different systems might affect different production performances, along with genetic responses, and could contribute to an alternative method for an animal breeding program under conditions typically experienced by farmers at smallholder farms. Therefore, the present study was conducted to examine the growth performance and genetics of Thai native chickens under conditions typically experienced by farmers at smallholder farms (on-site farm) compared with an experimental unit (on-station).

## 2. Materials and Methods

### 2.1. Research Sites

The study was carried out at the Research and Development Network Center of Animal Breeding and Omics, Khon Kaen University, Khon Kaen, Thailand. Meanwhile the farmers’ on-site experimental trials were performed in a rural community in Thung Pong Distinct, Khon Kaen Province (distance from on-station 50 km). These trials were supported by a career development initiative under the “Development of indigenous chicken farming for sustainability” project (reference no. 007/2562) through the collaboration between the Royal Initiative Discovery Foundation and the Research and Development Network Center of Animal Breeding and Omics, Khon Kaen University. This research project was reviewed and approved by the Institutional Animal Care and Use Committee of Khon Kaen University based on the Ethics of Animal Experimentation of the National Research Council of Thailand (No. IACUC-KKU-144/64).

### 2.2. Animal Management

On-station farm research.: A total of 613 chickens were produced in two hatches, randomly selected, and mixed by sex. After hatching, the chicks were weighed individually, and an identification number was attached to the leg. Chicken management occurred up to four weeks of age, followed by wing banding to maintain body weight and growth records. All chickens received Newcastle vaccines and antibiotics according to the chicken vaccination program.

On-site farm research. A total of 1081 chickens were produced in three hatches by farmers who had experienced raising native chickens before and had participated in training on raising native chickens with at Khon Kaen University. In addition, farmers must have had equipment for raising native chickens that included a chicken house with a concrete padded floor with rice husks foundation, while the four sides of the house were composed of wire mesh, with water tanks inside, feed tanks, and lighting. In front of the chicken house, there was a device to record body weight, feed intake, and other information. The chickens were raised within a house with 1 m^2^ per eight chickens. Therefore, each on-site farm received between 200–240 native chickens depending on the chicken house size. They were managed in a manner similar to that followed for the on-station research.

### 2.3. Animal Feeding

The chickens in both the on-site farm and on-station were fed with standard commercial broiler diets with two formulas according to the age of the chickens. From hatching to 4 weeks of age, 21% crude protein (CP) and 3000 Kcal/kg (Balance 910, Betagro company, Bangkok, Thailand) were provided, followed by a grower feed with 19% crude protein (CP), and 2900 Kcal/kg (Balance 911, Betagro company, Bangkok, Thailand) until the end of the experiment [27]. They were raised under the same environmental conditions with open-air housing. The chicks were raised with warming from a 100-watt lamp for four weeks. The lighting program consisted of two stages: the first stage was from hatching to 4 weeks with 24 h light/0 h dark, and the second stage was from after 4 to 16 weeks with 23 h light/1 h dark.

### 2.4. Data Collection

Individual chicken data consisted of the birth weight (BW0) (g), body weight at 4, 8, 12, and 16 weeks of age (BW4, BW8, BW12, and BW16, respectively (g)), average daily gain (g/day) during 0–4, 4–8, 8–12, and 12–16 weeks of age (ADG0–4, ADG4–8, ADG8–12, and ADG12–16, respectively), and breast circumference (cm) at 8, 12, and 16 (slaughter weight) weeks of age (BrC8, BrC12, and BrC16, respectively). Before using the data for statistical analysis, the Proc UNIVARIATE procedure by SAS v.9.0 software was used to examine data distribution, including assessing normality and checking data outliers.

### 2.5. Genetic Analysis

The recorded data were validated and analyzed for the least squares mean (LSmean) value, and significant differences were compared by research site (on-site farm and on-station) between mixed-sex and separated-sex chickens using the generalized linear model for an unbalanced analysis of variance (GLM procedure) in the SAS package to investigate significant differences. If significant differences were detected, then multiple pairwise comparisons were conducted using Scheffe’s test (*p* < 0.05).

The average information-restricted maximum likelihood (AI-REML) method with the best linear unbiased prediction (BLUP) were used to estimate the variance components, genetic parameters (heritability, genetic correlations, and phenotypic correlations), and estimated breeding values (EBVs) [28]. The multitrait animal model used in this study was as follows:Y=Xβ+Zα+ε 
where Y is the vector corresponding to the phenotypic values for the body weight and growth performance traits, namely, birth weight, body weight at 4, 8, 12 and 16 weeks of age (BW0, BW4, BW8, BW12, and BW16, respectively), ADG (g/day) from 0–4, 4–8, 8–12, and 12–16 weeks of age (ADG0–4, ADG4–8, ADG8–12, and ADG12–16, respectively), and breast circumference (cm) at 8, 12, and 16 (slaughtering weight) weeks of age (BrC8, BrC12, and BrC16, respectively). X and Z are the incidence matrices related to fixed and random effects, respectively. β is the vector of fixed effects, including the chicken hatch set and sex. α is the vector of random additive genetic effects, assumed to be $\alpha \;\sim N(0,A\sigma_{a}^{2}$), where A is an additive relationship matrix and $\sigma_{a}^{2}$ is the additive genetic variance, and ε is the vector of random residual effects, assumed to be $\varepsilon \;\sim N\left(0,I\sigma_{e}^{2}\right)$, where I is the identity matrix and $\sigma_{e}^{2}$ is the residual variance.

The generalized linear model for an unbalanced analysis of variance (PROC GLM) using the SAS package was used to compare the EBV by research site. The accuracy of the selection index by research site was calculated based on the following equation: $r=\sqrt{\frac{b^{\prime }Gb}{v^{\prime }Gv}}$, where $b=P^{-1}Gv$; *P* = phenotypic variance-covariance matrix, G = genetic variance-covariance matrix, with $G=A\sigma_{a}^{2},$ where A is an additive relationship matrix and $\sigma_{a}^{2}$ is the additive genetic variance, and v = the vector of relative economic weights corresponding to the traits considered in this study.

The selection index was calculated based on three traits: body weight, average daily gain, and breast circumference (used to represent the quantity of breast meat). The relative economic value (v) for each trait was calculated as a proportion of the standardized economic value to the total economic importance of all the traits evaluated in the given production system. We determined that both growth traits (body weight and average daily gain) and carcass traits (breast circumference) were of equal importance; therefore, the relative economic values were defined as 0.5 for growth traits and 0.5 for carcass traits. When considered in detail, the genetic correlations among body weight and ADG traits were large and positive; therefore, the relative economic values were assigned equal proportions of 0.25. The selection index equation is as follows:$$I=\left(v_{1}\times EBV_{BW..}\right)+\left(v_{2}\times EBV_{ADG..}\right)+\left(v_{3}\times EBV_{BrC..}\right)$$
where I is the selection index; $v_{1},v_{2},\;\mathrm{and}\;v_{3}\;$
are relative economic values for body weight, average daily gain, and breast circumference, respectively, and they have values of 0.25, 0.25, and 0.50, respectively.$EBV_{1},EBV_{2},\;\mathrm{and}\;EBV_{3}$ are estimates of the breeding values for the above traits, which correspond to the economic values.

## 3. Results

### 3.1. Growth Performance

A comparison of the least square means of the body weight, average daily gain, and breast circumference between Thai native chickens raised on-site farm and on-station is shown in Figure 1. The body weights at 8, 12, and 16 weeks of age (BW8, BW12, and BW16) in Figure 1A of the mixed-sex Thai native chickens raised on station (879.78, 1449.15, and 2029.10 g, respectively) were significantly higher than for those raised on farm (808.01, 1256.69, and 1604.71 g) (*p* < 0.05), while the birth weight and body weight at four weeks of age (BW0 and BW4) were not significantly different (*p* > 0.05). The results showed consistent trends with the sex-segregated analysis, as shown in Figure 1B,C. The average daily gain (ADG) traits (Figure 1D–F) were consistent with the body weight traits. In Figure 1D, the ADGs of the mixed-sex Thai native chickens raised on-station were higher than those raised on-site farm, especially the ADGs at 4–8, 8–12, and 12–16 weeks of age (18.54 vs. 15.90 g/day, 20.99 vs. 18.65 g/day, and 24.72 vs. 21.04 g/day, respectively), which were significantly different (*p* < 0.05). Only the ADG at 0–4 weeks of age was not significantly different between the on-site farm and on-station chickens (8.11 vs. 8.48 g/day) (*p* > 0.05). In terms of breast circumference (BrC) traits, we found significant differences at 8, 12, and 16 weeks of age (BrC8, BrC12, and BrC16) (*p* < 0.05) in both the mixed-sex and sex-segregated (male and female) chickens, as shown in Figure 1G–I. In the mixed-sex chickens, Thai native chickens raised on-station had breast circumferences of 24.25, 26.32, and 29.33 at 8, 12, and 16 weeks of age, respectively, which were higher than those raised on-site farm, which had breast circumferences of 21.02, 24.53, and 26.96 at 8, 12, and 16 weeks of age, respectively.

### 3.2. Estimated Heritability

Table 1 shows variance components and heritability of growth traits in term of body weight, average daily gain, and breast circumference at different ages in Thai native chicken raised on-site farm and on-station. The heritability estimates for body weight at all ages were moderate to high in the Thai native chickens raised on-site farm and on-station. The highest values were observed at birth, and the values decreased thereafter. In addition, the heritability estimates in Thai native chickens raised on-station were slightly higher than those raised on-site farm at all ages. The heritability estimates for the body weight of the Thai native chickens raised on-site farm at birth and 4, 8, 12, and 16 weeks of age (BW0, BW4, BW8, BW12, and BW16) were 0.499, 0.464, 0.394, 0.236, and 0.244, respectively, while the estimates for those raised on-station were 0.499, 0.496, 0.413, 0.317, and 0.291, respectively. The heritability estimates of ADG in Thai native chickens raised on-site farm and on-station at all ages were moderate, and ranged from 0.274 to 0.283 and 0.298 to 0.313, respectively. The highest heritability estimates of ADG were found from 0–8 weeks of age (ADG4–8) in both the on-site farm and on-station chickens. The heritability estimates of breast circumference at BrC8, BrC12, and BrC16 were moderate in both on-site farm and on-station, with slightly higher on-station values (0.296, 0.291, and 0.278, respectively) than on-site farm values (0.268, 0.251, and 0.204, respectively).

### 3.3. Phenotypic and Genetic Correlation

The phenotypic correlations of body weight (BW), average daily gain (ADG), and breast circumference (BrC) in Thai native chickens raised on-site farm and on-station are presented in Table 2. The results showed positive phenotypic correlations among and between all growth traits, and they ranged from low to high values (0.02 to 0.90). In addition, the phenotypic correlations among and between traits were slightly higher in Thai native chickens raised on-site farm (0.10 to 0.90) than in those raised on-station (0.02 to 0.80). For body weight, the phenotypic correlation between birth weight (BW0) and other body weights at different ages (BW4, BW8, BW12, BW16) in Thai native chickens raised both on-site farm and on-station was lowest (<0.13) when compared to the phenotypic correlation among body weights at different ages. In addition, we found high phenotypic correlations among body weight traits at eight weeks of age (BW8) and for body weight at 12 and 16 (slaughtering weight) weeks of age (0.76 and 0.65). The phenotypic correlations between BW8 and ADG (0.61 to 0.89) and BW8 and BrC (0.47 to 0.79) at different ages were also moderate to high in Thai native chickens raised on-site farm and on-station. In terms of average daily gain, the phenotypic correlations among and between traits appeared to be numerically similar (moderate to high) to the phenotypic correlations among and between body weight traits; however, the correlations between similar traits, such as between ADG0–4 and ADG4–8 (0.63 and 0.74), were higher than those between more dissimilar traits, such as between ADG0–4 and ADG8–12 (0.50 and 0.62) or between ADG0–4 and ADG12–16 (0.44 and 0.52).

The genetic correlations of body weight (BW), average daily gain (ADG), and breast circumference (BrC) in Thai native chickens raised on-site farm and on-station are presented in Table 3. The genetic correlations among and between growth traits in chickens raised on-site farm (0.04 to 0.96) were slightly lower than for those raised on-station (0.04 to 0.98). The genetic correlations among the body weight traits (BW0, BW4, BW8, BW12, and BW16) in on-site farm and on-station chickens were positive and ranged from low to high, varying from 0.09 to 0.73 for on-site farm chickens and from 0.11 to 0.81 for on-station chickens. The genetic correlations among the ADG traits (ADG0–4, ADG4–8, ADG8–12, and ADG12–16) for the on-site farm and on-station chickens were positive and had medium values varying from 0.46 to 0.74 for the on-site farm chickens, and from 0.58 to 0.79 for the on-station chickens. Additionally, the genetic correlations among the breast circumference traits (BrC8, BrC12, and BrC16) were medium, with the values ranging from 0.35 to 0.48 for on-site farm and from 0.42 to 0.62 for on-station chickens. The genetic correlations between body weight and ADG ranged from low to high, and they presented the same trends as the genetic correlations between body weight and breast circumference and the genetic correlations between ADG and breast circumference.

### 3.4. Estimated Breeding Value and Selection Index

The top 20% of the estimated breeding values and selection indices for the on-site farm and on-station Thai native chickens are presented in Figure 2. The selection of animals for replacement herds was based on the estimated breeding values (EBVs), with each generation selecting the first 20% of the best EBVs. Significant differences in the EBVs for body weight, average daily gain, and breast circumference traits were found at eight weeks of age, with the on-station EBVs higher than on-site farm EBVs. At eight weeks of age, the EBVs of body weight were 53.14 g and 48.72 g for the on-station and on-site farm chickens, respectively. The EBVs of ADG were 1.52 g/day and 1.27 g/day for the on-station and on-site farm chickens, respectively, and the EBVs of breast circumference were 0.32 cm and 0.26 cm for on-site farm for the on-station and on-site farm chickens, respectively. The appropriate selection index in this study consisted of three traits: body weight at eight weeks of age, ADG from 4–8 weeks of age, and breast circumference at eight weeks of age. The results showed that the selection index of chickens raised on station (13.85) was higher than that of those raised on-site farm (9.66). Moreover, the accuracy of the selection index was 10.67% (0.77 and 0.68) greater for the on-station chickens than for the on-site farm chickens.

## 4. Discussion

This study used a multitrait animal model and selection index to compare the growth performance and genetic variations in Thai native chickens raised in different environments, namely, on-site farm and on-station environments. Differences were not observed in birth weight, body weight at four weeks of age or ADG from 0–4 weeks of age in the mixed-sex and sex-segregated chickens raised on-site farm and on-station (*p* > 0.05). However, from 8 weeks onward, the native chickens raised on-station had statistically significantly higher body weight than the native chickens raised on-site farm (*p* < 0.05). The differences in body weight between the native chickens raised on-station and on-site farm were 72, 192, and 424 g at 8, 12, and 16 weeks, respectively. The results of this study agree with several studies. For example, Lwelamira [23], Tanzanian native chickens raised at the testing station at eight weeks of age had higher body weight than those raised on the farm. At the same time, Guni et al. [24] found that at 20 weeks of age, Kuroiler and Sasso chickens (exotic breeds) raised in the testing station had higher body weights (2313.1 and 2708.8 g, respectively) than those raised on the farm (1995.2 and 1745.9 g, respectively). However, a comparison of the body weight of the native chickens in this study with that of other Thai native chickens showed that the chickens in this study had body weights 122 g higher than that of Chee chickens [9] and 88 g higher than that of Leung Hang Khao chickens [13,14]. In addition, the body weights were higher than those of native chickens in Korea [25], chicken strains in Egypt [26], Venda chickens of South Africa [29], and native chickens of Ethiopia [30]. These results indicate that this study’s method of genetic improvement of Thai native chickens is valid. In other words, the selection by genetic evaluation method and selected chickens from 8 weeks of age allows for faster selection progression.

The low body weight in native chickens raised on-site farms may be due to different management factors, such as feed shortages in quantity and insufficient nutrients supplied for growth, and the high prevalence of diseases and parasites that usually prevail under such a system [31,32,33]. In addition, differences between on-station and on-site farm management were observed and contributed to the differences in growth characteristics, since the lighting, ventilation system, chicken stocking density, and bedding materials are management factors that were found to be involved in differences in growth characteristics on both farms, particularly those on-site farms, were found to be insufficient or absent at the farm. Research indicated that light (in terms of the light source and period) was a crucial environmental factor affecting poultry performance, immune response, livability, and health status [34,35]. At the same time, the air velocity by natural ventilation may not be sufficient to reduce the air pollution of the chicken house, such as ammonia, dust, and other gases, which can influence production performances and the chickens’ health [36]. Therefore, using exhaust fans, especially in hot weather, can maintain chickens’ health and improve productivity [37]. Meanwhile, several studies have been conducted to study the effect of stocking density on chicken production and performance, and showed the benefits of reducing stocking density on the performance of broilers [38,39]. Abudabos et al. [40] concluded that increasing the stocking density rate from 28 to 40 kg of BW/m^2^ had impingement effects on broiler chicken performance and could jeopardize their welfare. For bedding, it was found that there was not much difference between sawdust and rice husk [41]; however, choosing a material readily available in the local area was probably the best option in terms of handling.

One of the things that support differences in growth characteristics in both farms is that when we finished the experiment, we interviewed the Pradu Hang dum chicken farmers in the experimental area (on-site farms) about their satisfaction. We received interesting information that some farmers had other activities besides raising chickens, such as rice planting and cassava farming. Therefore, it may be one reason farmers do not fully spend time raising native chickens [42,43].

The heritability estimates for the BW, ADG, and BrC traits in the present study were medium to high (ranging from 0.204 to 0.499; see Table 1), which was similar to the results of studies carried out on local Venda chickens [29], Mazandaran native chickens [44], and purebred [13,14] and crossbred Thai native chickens [45,46]. The heritability for body weight was highest at eight weeks of age (BW8), not including birth weight (BW0), and the values tended to decrease with increasing age. Similar results were observed in Horro chickens in Ethiopia [47], native chickens in Korea [48] and another breed of native chicken in Thailand [45,46]. Heritability helps in decision-making in farm management strategies, estimating breeding values, predicting genetic progress under selection, and even in predicting expected production in the future [49]. If the studied trait has a moderate-to-high heritability value, then the trait can be improved by focusing on improving the genetic conditions, which is more cost-effective than improving the environment. In addition, if the traits have a high heritability value, then genetic improvements result in very quick and accurate selection. However, the clear difference between this study and previous studies is that the native chickens from this study are continuously genetically developed before being tested in the on-site farms. Data on differences in growth performance from different raising areas were obtained, and genetic comparisons of individual chickens from both areas were also obtained, which had not been conducted in previous studies. This finding demonstrated that Thai native chickens raised in areas with different management conditions, in addition to having different growth performance, also had different genetic expression.

The phenotypic correlations (on-site farm and on-station) between the growth traits were positive and ranged from 0.02 to 0.90 (see Table 2). The phenotypic correlations in Thai native chickens raised in the on-site farm was higher than those in Thai native chickens raised on-station, indicating that the environment had a greater influence on the phenotype of traits in on-site farm raised chickens than on-station raised chickens. If the influence of the environment is sufficiently larger than that of genetic factors, then the genetic expression of the trait can also be altered, which is known as the genotype-environment interaction [50]. The low genetic correlations (on-site farm and on-station) between BW0 and other traits indicated that this trait could not be used in the genetic selection of Thai native chickens in this study (see Table 3), which is also associated with the low selection accuracy. In addition, the genetic correlations between BW4 and other traits were suitable for body weight only but presented low values with average daily and breast circumference traits. The genetic correlations between BW8 and other traits were strong and positive, especially with the ADG from 4–8 weeks of age (0.89 and 0.94) and the breast circumference at eight weeks of age (0.60 and 0.67). Moreover, BW8 also had higher genetic correlations with BW12 (0.73 and 0.77) and BW16 (0.63 and 0.69) (slaughter weight) than BW4. Positive genetic correlations indicated that selection of one trait can improve the performance of other traits. These results were consistent with previous studies, such as Thai native Leung Hang Khao chickens [14], crossbred Thai native chickens [45,46], four Egyptian chicken strains [26], and Mazandaran native chickens [44].

The EBV showed that the native chickens raised on-station presented high phenotype growth efficiency (see Figure 2) and higher genetic growth efficiency than the native chickens raised on-site farm. Using EBVs to select animals is more efficient than phenotypic selection because the animals were directly selected according to their genetic value [51]. Moreover, the growth traits body weight, average daily gain, and breast circumference could be selected simultaneously. Regarding the accuracy of the selection index (Figure 2), we found that the accuracy value of on-station chickens was greater than that of on-site farm chickens (0.77 and 0.68, respectively). It might be inferred that the data from on-station farm research were reliable because of the amount of data and data connectedness [52,53], the multiple sources of data [54], and the high heritability traits [55]. In addition, other studies showed positive EBVs of body weight and found that these values tended to increase with selection [56,57,58]. Therefore, we inferred that a multitrait animal model and a selection index approach could be used for accurate genetic selection in both on-site farm and on-station chickens.

## 5. Conclusions

The results of this study indicate that different environmental effects (on-site farm and on-station) affect both the growth characteristics and genetic expression of Thai native chickens. The phenotypic and genetic growth traits of native chickens raised on-station showed better correlations than those of native chickens raised in an on-site farm. In addition, the genetic parameters and breeding values in on-station showed that the breeding program by selection index for improving growth performance is valid. Therefore, to implement such a breeding program in an on-site farm, an intensive or semi-intensive farm system should be considered to minimize the effect of genotype-environment interaction.

## Figures and Tables

**Figure 1 vetsci-10-00011-f001:**
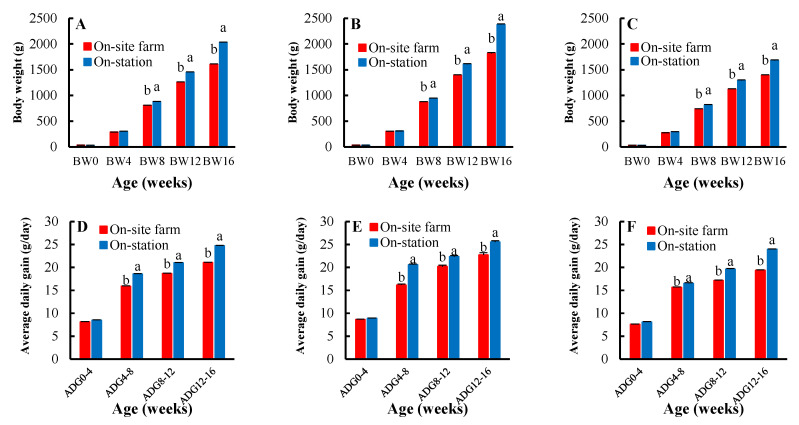
Least square means ± standard errors of the body weight of (**A**) mixed-sex chickens, (**B**) male chickens, and (**C**) female chickens at birth and 4, 8, 12, 16 weeks of age (BW0, BW4, BW8, BW12, BW16); average daily gain of (**D**) mixed-sex chickens, (**E**) male chickens, and (**F**) female chickens from 0–4, 4–8, 8–12, and 12–16 weeks of age (ADG0–4, ADG4–8, ADG8–12, ADG12–16); breast circumference of (**G**) mixed-sex chickens, (**H**) male chickens, and (**I**) female chickens at 8, 12 and 16 weeks of age (BrC8, BrC12, BrC16) between the on-site farm (red bar) and on-station (blue bar) experiments using Thai native chickens (Pradu Hang dum). Means for the same trait with different letters (a, b) differ significantly (*p* < 0.05).

**Figure 2 vetsci-10-00011-f002:**
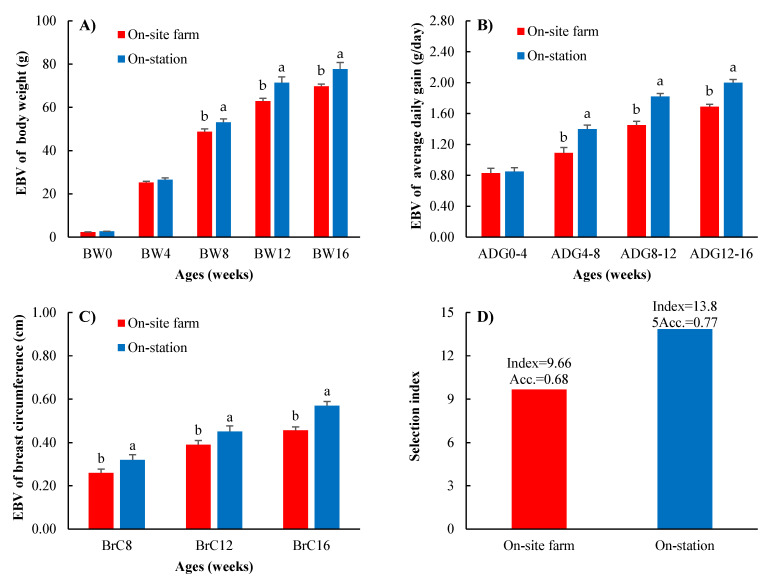
Top 20% estimated breeding values of (**A**) body weight, (**B**) average daily gain, (**C**) breast circumference, (**D**) selection index (Index) and accuracy of selection index (Acc.) between on-site farm (red bar) and on-station (blue bar) Thai native chickens (Pradu Hang dum). Means for the same trait with different letters (a, b) differ significantly (*p* < 0.05).

**Table 1 vetsci-10-00011-t001:** Variance components and heritability (±standard error; SE) of growth traits between the on-site farm and on-station Thai native chickens (Pradu Hang dum).

Research Sites	Traits	Parameters		
		$\sigma_{a}^{2}$	$\sigma_{e}^{2}$	$h^{2}\left(SE\right)$
On-site farm	BW0	5.26	5.28	0.499 ± 0.001
	BW4	632.21	729.15	0.464 ± 0.002
	BW8	3055.07	4691.28	0.394 ± 0.005
	BW12	4199.35	13,560.11	0.236 ± 0.007
	BW16	8541.00	26,410.00	0.244 ± 0.007
	ADG0–4	0.81	2.12	0.276 ± 0.002
	ADG4–8	1.01	2.56	0.283 ± 0.003
	ADG8–12	1.60	4.10	0.281 ± 0.009
	ADG12–16	1.95	5.17	0.274 ± 0.015
	BrC8	0.30	0.82	0.268 ± 0.013
	BrC12	0.42	1.25	0.251 ± 0.021
	BrC16	0.46	1.79	0.204 ± 0.004
On-station	BW0	3.21	3.22	0.499 ± 0.001
	BW4	746.00	757.20	0.496 ± 0.001
	BW8	3143.15	4474.13	0.413 ± 0.001
	BW12	5422.22	11,675.41	0.317 ± 0.002
	BW16	9245.42	22,480.20	0.291 ± 0.005
	ADG0–4	0.85	2.00	0.298 ± 0.033
	ADG4–8	1.18	2.59	0.313 ± 0.046
	ADG8–12	1.85	4.09	0.311 ± 0.008
	ADG12–16	2.30	5.22	0.306 ± 0.021
	BrC8	0.40	0.95	0.296 ± 0.021
	BrC12	0.55	1.12	0.291 ± 0.016
	BrC16	0.64	1.66	0.278 ± 0.011

On-site farm = farmer’s farms; On-station = Khon Kaen University experimental farm; BW0, BW4, BW8, BW12, and BW16 = birth weight, body weight at 4, 8, 12, 14 and 16 weeks of age, respectively; ADG0–4, ADG4–8, ADG8–12, and ADG12–16 = average daily gain (g/day) during 0–4, 4–8, 8–12, and 12–16 weeks of age, respectively; BrC8, BrC12, and BrC16 = breast circumference (cm) at 8, 12, and 16 weeks of age, respectively; $\sigma_{a}^{2}=$
additive genetic variances; $\sigma_{e}^{2}=$
residual variances; $h^{2}=$
heritability.

**Table 2 vetsci-10-00011-t002:** Phenotypic correlations of growth traits between on-site farm (above diagonal) and on-station (below diagonal) Thai native chickens (Pradu Hang dum).

Traits	BW0	BW4	BW8	BW12	BW16	ADG0–4	ADG4–8	ADG8–12	ADG12–16	BrC8	BrC12	BrC16
BW0	-	0.12	0.10	0.11	0.12	0.15	0.13	0.11	0.11	0.13	0.09	0.06
BW4	0.09	-	0.76 *	0.57 *	0.48	0.80 *	0.78 *	0.60 *	0.49 *	0.52 *	0.47 *	0.34
BW8	0.13	0.65 *	-	0.84 *	0.76 *	0.75 *	0.89 *	0.84 *	0.79 *	0.79 *	0.68 *	0.56 *
BW12	0.10	0.49 *	0.74 *	-	0.87 *	0.59 *	0.86 *	0.88 *	0.88 *	0.57 *	0.78 *	0.74 *
BW16	0.13	0.43	0.63 *	0.73 *	-	0.50	0.77 *	0.86 *	0.90 *	0.54 *	0.66 *	0.83 *
ADG0–4	0.06	0.70 *	0.66 *	0.50 *	0.44	-	0.74 *	0.60 *	0.52 *	0.55 *	0.48 *	0.45 *
ADG4–8	0.12	0.66 *	0.80 *	0.76 *	0.65 *	0.63 *	-	0.82 *	0.74 *	0.80 *	0.70 *	0.67 *
ADG8–12	0.10	0.52 *	0.78 *	0.80 *	0.68 *	0.50 *	0.74 *	-	0.79 *	0.628 *	0.79 *	0.75 *
ADG12–16	0.10	0.46	0.61 *	0.62 *	0.81 *	0.44 *	0.60 *	0.64 *	-	0.56 *	0.68 *	0.84 *
BrC8	0.07	0.47 *	0.65 *	0.56 *	0.46 *	0.50 *	0.69 *	0.59 *	0.46 *	-	0.59 *	0.55 *
BrC12	0.03	0.41 *	0.59 *	0.64 *	0.45 *	0.45 *	0.61 *	0.66 *	0.46 *	0.49 *	-	0.60 *
BrC16	0.02	0.36	0.47 *	0.45 *	0.69 *	0.40 *	0.50 *	0.48 *	0.72 *	0.39 *	0.41 *	-

On-site farm = farmer’s farm; On-station = Khon Kaen University experimental farm; BW0, BW4, BW8, BW12, and BW16 = birth weight, body weight at 4, 8, 12, 14 and 16 weeks of age (g); ADG0–4, ADG4–8, ADG8–12, and ADG12–16 = average daily gain during 0–4, 4–8, 8–12, and 12–16 weeks of age (g/day); BrC8, BrC12, and BrC16 = breast circumference at 8, 12, and 16 weeks of age (cm); * indicates a significant value (*p* < 0.05).

**Table 3 vetsci-10-00011-t003:** Genetic correlations of growth traits between on-site farm (above diagonal) and on-station (below diagonal) Thai native chickens (Pradu Hang dum).

Traits	BW0	BW4	BW8	BW12	BW16	ADG0–4	ADG4–8	ADG8–12	ADG12–16	BrC8	BrC12	BrC16
BW0	-	0.09	0.13	0.09	0.12	0.04	0.10	0.06	0.05	0.10	0.07	0.02
BW4	0.20	-	0.65 *	0.49 *	0.44	0.79 *	0.64 *	0.48 *	0.38	0.35 *	0.32	0.29
BW8	0.16	0.74 *	-	0.73 *	0.63 *	0.65 *	0.89 *	0.76 *	0.54 *	0.60 *	0.54 *	0.38 *
BW12	0.11	0.53 *	0.77 *	-	0.62 *	0.50 *	0.74 *	0.96 *	0.58 *	0.65 *	0.62 *	0.35 *
BW16	0.12	0.46	0.69 *	0.81 *	-	0.42	0.60 *	0.62 *	0.97 *	0.45 *	0.48 *	0.57 *
ADG0–4	0.15	0.90 *	0.77 *	0.52 *	0.42	-	0.65 *	0.50 *	0.46 *	0.57 *	0.42 *	0.40 *
ADG4–8	0.12	0.75 *	0.94 *	0.74 *	0.64 *	0.74 *	-	0.74 *	0.61 *	0.66 *	0.59 *	0.50 *
ADG8–12	0.06	0.50 *	0.77 *	0.97 *	0.75 *	0.60 *	0.79 *	-	0.59 *	0.74 *	0.66 *	0.47 *
ADG12–16	0.07	0.40	0.70 *	0.85 *	0.98 *	0.58 *	0.72 *	0.75 *	-	0.58 *	0.45 *	0.68 *
BrC8	0.08	0.43 *	0.67 *	0.72 *	0.45 *	0.64 *	0.68 *	0.80 *	0.65 *	-	0.40 *	0.35 *
BrC12	0.04	0.30 *	0.50 *	0.60 *	0.44 *	0.50 *	0.67 *	0.72 *	0.60 *	0.42 *	-	0.48 *
BrC16	0.04	0.21	0.38 *	0.43 *	0.52 *	0.50 *	0.62 *	0.64 *	0.75 *	0.44 *	0.62 *	-

On-site farm = farmer’s farm; On-station = Khon Kaen University experimental farm; BW0, BW4, BW8, BW12, and BW16 = birth weight, body weight at 4, 8, 12, 14 and 16 weeks of age (g); ADG0–4, ADG4–8, ADG8–12, and ADG12–16 = average daily gain during 0–4, 4–8, 8–12, and 12–16 weeks of age (g/day); BrC8, BrC12, and BrC16 = breast circumference at 8, 12, and 16 weeks of age (cm); * indicates a significant value (*p* < 0.05).

## Data Availability

The data presented in this study are available on request from the Network Center for Animal Breeding and Omics Research, Faculty of Agriculture, Khon Kaen University, Thailand.

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
