# Peer review of "Genetic Comparisons of Body Weight, Average Daily Gain, and Breast Circumference between Slow-Growing Thai Native Chickens (Pradu Hang dum) Raised On-Site Farm and On-Station"

_vetsci, 2022, doi:10.3390/vetsci10010011_

Round 1
Reviewer 1 Report
· Suitability for the journal. I wonder whether this manuscript fits the aims and scopes of Vet. Sci. since health/veterinary aspects are not part of the study. The information given on the journal website are quite clear; articles should be relevant to any field of veterinary sciences "..., including prevention, diagnosis and treatment of disease, disorder and injury in animals". None of these topics is addressed in the present manuscript.
· Line 11-13. Revise sentence wording.
· Line 32-36. Well noted, but what was the benefit from this study. Surely, earlier studies on chicken breeding with different genotypes are available. Were the results so different or in other terms, did authors expect for whatever reason that Thai native chickens follow other rules of artificial selection and quantitative genetics, respectively?
· Line 47. Here and elsewhere, do not refer to animal nutrition as "food" but "feed".
· Lines 48-49. This appears to be a somewhat superficial statement. "Native chicken" is not a breed per se. Shouldn't such abilities been specific for certain genotypes instead of an extensive chicken production system?
· Lines 50-51. Again, does this conclusion also relate to the rest of the world?
· Lines 54-63. The slower growth rate obviously affects feed conversion. This should be addressed in the introduction.
· Lines 75-82. Authors mentioned earlier that benefits of "Native Chickens" include better product quality in terms of taste and nutrients, easy rearing abilities, disease resistance etc. compared to the high performing breeds. These are all functional traits and it is well accepted that breeding for higher performance will impair such. I therefore wonder what the benefit is from establishing such breeding programs since it may lead for respective breeds like Prad Hang Dum to get more alike with high performers like ROSS, thereby rendering them potentially unsuitable for “backyard rearing”.
· Line 88. I wonder if the sample size is high enough to estimate parameters of quantitative genetics. Usually, sample size are much larger especially with respect to chicken breeding.
· Instructions for authors. Manuscript does not meet author guidelines. E.g. SI units were not used in some cases (e.g. calorie vs. joule).
· Lines 100-103. It must be highlighted which feeding recommendations were followed. Since I assume that commercial diets were used, respective information (feed labels) should be added as supplementary material.
Author Response
Dear Reviewer,
We are grateful for the critical reading and your efforts to improve the quality of the manuscript. For the questions and comments, we have responded to each of you individually according to the attached file and as shown below. We hope that the manuscript in its revised form will please you.
Response to Reviewer 1 Comments
Point 1: Suitability for the journal. I wonder whether this manuscript fits the aims and scopes of Vet. Sci. since health/veterinary aspects are not part of the study. The information given on the journal website are quite clear; articles should be relevant to any field of veterinary sciences "..., including prevention, diagnosis and treatment of disease, disorder and injury in animals". None of these topics is addressed in the present manuscript.
Response 1: Actually, genetics is one of the subject areas of Veterinary Sciences which is indicated in the Scope of the journal. The study of genetics in indigenous chickens raised in different fields (on-farm and on-station) and their growth performance have been reported in our study. The results of such investigations could be used to support and genetically improve chickens to promote their husbandry as global food as increasing demand nowadays.
Point 2: Line 11-13. Revise sentence wording.
Response 2: we have revised more validation as follows and see lines 11-16:
Former sentence: Growth characteristics are directly essential for and the focus of poultry production; therefore, the current study aimed to compare the growth performance and genetics of native chickens raised on farmers’ farms (on-farm) and the Khon Kaen University experimental farm (on-station).
Revised sentence: To ensure that any new technology developed within the experimental station is appropriate to the community’s needs and compatible with the existing systems, on-site farm research is an important component in examining the effectiveness of agricultural research. The present study examined the growth performance and genetics of Thai native chickens under conditions typically experienced by farmers at smallholder farms (on-site farm) compared with an experimental unit (on-station).
Point 3: Line 32-36. Well noted, but what was the benefit from this study. Surely, earlier studies on chicken breeding with different genotypes are available. Were the results so different or in other terms, did authors expect for whatever reason that Thai native chickens follow other rules of artificial selection and quantitative genetics, respectively?
Response 3: In this study, we have developed new chicken genetics along with tested them for genetic performance on the farmer’s farm. By doing so, we can get chickens that have both good growth performance and genetic ability that truly meet the needs of farmers.
Unlike previous study, it was only comparing the chicken genetics that farmers had already raised in the area. Therefore, the answer that farmers received was only the chicken breed that should be raised in the area without the chicken has not been genetic improvement in anyway.
However, to make the conclusion of this study clearer, the authors have added a conclusion sentence as follows: “In conclusion, the genetic parameters and breeding values in on-station showed that the breeding program by selection index for improving growth performance is valid. Therefore, to implement such a breeding program in on-site farm, the intensive or semi-intensive farm system should be considered to minimize the effect of genotype-environment interaction.” Please see in lines 39-42.
Point 4: Line 47. Here and elsewhere, do not refer to animal nutrition as "food" but "feed".
Response 4: the “food” was changed to “feed” throughout the revised MS as the reviewer’s suggestion. See line 50.
Point 5: Lines 48-49. This appears to be a somewhat superficial statement. "Native chicken" is not a breed per se. Shouldn't such abilities been specific for certain genotypes instead of an extensive chicken production system?
Response 5: we have revised the mentioned sentence with more specificity as follows and see lines 51-56.
Former sentence: Native chicken meat shows good flavor, tenderness, and low fat content [13-15], and it contains more vital nutrients for consumers than commercial broilers and less fat than pork and cattle [16-18].
Revised sentence: From a marketing perspective, native chicken meat has been proven to stand out in many respects, such as its distinct taste and flavor [7], and used as a health food as its low cholesterol [8] but high nutritional components [9-11], which are essential human health considerations. Therefore, native chicken meat has created a rapidly increasing demand in the chicken market for consumers [12].
Point 6: Lines 50-51. Again, does this conclusion also relate to the rest of the world?
Response 6: we have considered and designed to delete that reductant sentence without affecting the MS.
Point 7: Lines 54-63. The slower growth rate obviously affects feed conversion. This should be addressed in the introduction.
Response 7: we have added this sentence in the revised MS as the reviewer’s suggestion. See lines 59-60.
Point 8: Lines 75-82. Authors mentioned earlier that benefits of "Native Chickens" include better product quality in terms of taste and nutrients, easy rearing abilities, disease resistance etc. compared to the high performing breeds. These are all functional traits and it is well accepted that breeding for higher performance will impair such. Therefore, I wonder what the benefit is from establishing such breeding programs since it may lead to respective breeds like Prad Hang Dum to get along with high performers like ROSS, thereby rendering them potentially unsuitable for “backyard rearing”.
Response 8: Its truth, as the reviewer mentioned on the benefits of native chickens in many respects, such as their distinct taste, flavor, and more vital nutrients; that is why Chicken meat consumption continues to grow and remains universally popular among consumers. However, the non-comparable of native chickens with commercials is on their production (growth performance), even those that the animal breeders have developed. Also, the raising systems between commercial and native chickens are totally different. The Ross chickens are fast-growing broilers and raised in an evaporative cooling system, a poultry house with evaporative cooling pads for ventilating and cooling the house to a suitable ambient temperature (15–25 °C), while the latter chickens are heat tolerance, so they are easy to raise and well adapted to local conditions as “backyard rearing.”
Therefore, for the present study, to ensure that “Pradu Hang dum KKU55” which has been genetically improved in its growth performance after seven generations within our experimental station is appropriate to the community’s needs and compatible with the existing systems, on-site farm (smallholder farmers) is an important component in examining the effectiveness of breeding program. In the past, several studies compared the effects of farming systems (intensive and extensive) on genetics parameters [23-26], however, the study on the genetics of each individual chicken which is the ultimate goal for genetic selection, has not been investigated. It seems possible that different systems might affect different production performances along with genetic responses and could contribute to the alternative method for the animal breeding program under conditions typically experienced by farmers at smallholder farms. Therefore, the present study was conducted to examine the growth performance and genetics of Thai native chickens under conditions typically experienced by farmers at smallholder farms (on-site farm) compared with an experimental unit (on-station).
We apologize for the unclear of our writing. For more clarity on this topic, we have revised the introduction part in the revised MS. Please see lines 81-94.
Point 9: Line 88. I wonder if the sample size is high enough to estimate parameters of quantitative genetics. Usually, sample size are much larger especially with respect to chicken breeding.
Response 9: It’s true that a large amount of data is essential for the estimation of quantitative genetic parameters. However, the results in our study were comparable with the previous studies with large data (see reference list below). Besides, the standard error (SE) is smaller which indicates that the estimated value is exactly the true value.
Therefore, we believe that our results were reliable and the data in our study was enough to allow the efficient prediction of breeding values.
Reference list:
- Chomchuen, K.; Tuntiyasawasdikul, V.; Chankitisakul, V.; Boonkum, W. Comparative study of phenotypes and genetics related to the growth performance of crossbred Thai indigenous (KKU1 vs. KKU2) chickens under hot and humid conditions. Vet Sci. 2022, 9,
- Guni, F.S.; Mbaga, S.H.; Katule, A.M. Performance evaluation of Kuroiler and Sasso chicken breeds reared under on-farm and on-station management conditions in Tanzania. J. Sci. Food. Agric. 2021, 3, 53-59.
- Cahyadi, M.; Park, H-B.; Seo, D-W.; Jin, S.; Choi, N.; Heo, K-N.; Kang, B-S.; Jo, C.; Lee, J-H. Genetic parameters for growth-related traits in Korean native chicken. J. Poult. Sci. 2015, 42, 285-289.
- Lwelamira, J. Genotype-environmental (G x E) interaction for body weights for Kuchi chicken ecotype of Tanzania reared on-station and on-farm. J. Poult. Sci. 2012, 11, 96-102
- Norris, D.; Ngambi, J.W. Genetic parameter estimates for body weight in local Venda chickens. Anim. Health. Prod. 2006, 38, 605-609.
Point 10: Instructions for authors. Manuscript does not meet author guidelines. E.g. SI units were not used in some cases (e.g. calorie vs. joule).
Response 10: we have carefully rechecked and edited as the author’s guidelines.
Point 11: Lines 100-103. It must be highlighted which feeding recommendations were followed. Since I assume that commercial diets were used, respective information (feed labels) should be added as supplementary material.
Response 11: the commercial diets were used in the present study; we indicate this clearly in the text. See lines 131-136.
Best regards,
Dr.Wuttigrai Boonkum
Corresponding author

Reviewer 2 Report
1- English should improve by a native person. The paper suffers from a poor English structure throughout and cannot be published or reviewed properly in the current format. The manuscript requires a thorough proofread by a native person whose first language is English. The instances of the problem are numerous and this reviewer cannot individually mention them. It is the responsibility of the author(s) to present their work in an acceptable format. Unless the paper is in a reasonable format, it should not have been submitted.
2- The novelty of the study needs to be highlighted compare to other similar studies or consider to explicitly mention what is gap knowledge and/or what was lacking in the indicated studies.
3- Discussion is weak. The discussion needs enhancement with real explanations not only agreements and disagreements. Authors should improve it by the demonstration of causes of obtained results. Instead of just justifying results, results should be interpreted, explained to appropriately elaborate inferences. Discussion seems to be poor, didn't give good explanations of the results obtained. I think that it must be really improved. Where possible please discuss potential mechanisms behind your observations. You should also expand the links with prior publications in the area, but try to be careful to not over-reach. For the latter, you should highlight potential areas of future study.
4- The scientific background of the topic is poor. In "Introduction" and "Discussion", the authors should cite recent references between 2020-2022 from JCR journals (with impact factor) about recent achievements on the slow-growing chicken. For example, authors should cite to:
Singh, M., Lim, A. J., Muir, W. I., & Groves, P. J. (2021). Comparison of performance and carcass composition of a novel slow-growing crossbred broiler with fast-growing broiler for chicken meat in Australia. Poultry Science, 100(3), 100966.
Petkov E., Ignatova M., Popova T. and Ivanova S. (2020). Quality of eggs and hatching traits in two slow-growing dual-purpose chicken lines reared conventionally or on pasture. Iranian J. Appl. Anim. Sci. 10(1), 141-148.
de Jong, I. C., Blaauw, X. E., van der Eijk, J. A., da Silva, C. S., van Krimpen, M. M., Molenaar, R., & van den Brand, H. (2021). Providing environmental enrichments affects activity and performance, but not leg health in fast-and slower-growing broiler chickens. Applied Animal Behaviour Science, 241, 105375.
5- A detailed "Conclusion" should be provided to state the final result that the authors have reached. Please note you only need to place your conclusion and not keep putting results, because these have already been presented in the manuscript.
6- Author(s) should re-format the references based on journal format. See the instructions for authors.
7- The numbers and decimals in Tables should be follow the rule of: xxxx, xxx, xx.x, x.xx, 0.xxx and 0.0xxx
Author Response
Dear Reviewer,
We are grateful for the critical reading and your efforts to improve the quality of the manuscript. For the questions and comments, we have responded to each of you individually according to the attached file and as shown below. We hope that the manuscript in its revised form will please you.
Response to Reviewer 2 Comments
Point 1: English should improve by a native person. The paper suffers from a poor English structure throughout and cannot be published or reviewed properly in the current format. The manuscript requires a thorough proofread by a native person whose first language is English. The instances of the problem are numerous and this reviewer cannot individually mention them. It is the responsibility of the author(s) to present their work in an acceptable format. Unless the paper is in a reasonable format, it should not have been submitted.
Response 1: Actually, the manuscript has been corrected for spelling, grammar, and punctuation by Elsevier Language Editing Services which guarantees to edit by native speakers as per the attached certificate form. However, we have carefully edited the entire MS to improve clarity as the reviewer’s suggestion.
Point 2: The novelty of the study needs to be highlighted and compare to other similar studies or consider to explicitly mention what is gap knowledge and/or what was lacking in the indicated studies.
Response 2: In the present study, to ensure that “Pradu Hang dum KKU55” which has been genetically improved in its growth performance after seven generations within our experimental station is appropriate to the community’s needs and compatible with the existing systems, on-site farm (smallholder farmers) is an important component in examining the effectiveness of breeding program. In the past, several studies compared the effects of farming systems (intensive and extensive) on genetics parameters [23-26], however, the study on the genetics of each individual chicken which is the ultimate goal for genetic selection, has not been investigated. It seems possible that different systems might affect different production performances along with genetic responses and could contribute to the alternative method for the animal breeding program under conditions typically experienced by farmers at smallholder farms. Therefore, the present study was conducted to examine the growth performance and genetics of Thai native chickens under conditions typically experienced by farmers at smallholder farms (on-site farm) compared with an experimental unit (on-station).
We apologize for the unclear of our writing. For more clarity on this topic, we have revised the introduction part in the revised MS. Please see lines 81-94.
Point 3: Discussion is weak. The discussion needs enhancement with real explanations not only agreements and disagreements. Authors should improve it by the demonstration of causes of obtained results. Instead of just justifying results, results should be interpreted, explained to appropriately elaborate inferences. Discussion seems to be poor, didn't give good explanations of the results obtained. I think that it must be really improved. Where possible please discuss potential mechanisms behind your observations. You should also expand the links with prior publications in the area, but try to be careful to not over-reach. For the latter, you should highlight potential areas of future study.
Response 3: we have revised the discussion as the reviewer’s suggestion. See the highlight text of the discussion session in the revised MS.
Point 4: The scientific background of the topic is poor. In "Introduction" and "Discussion", the authors should cite recent references between 2020-2022 from JCR journals (with impact factor) about recent achievements on the slow-growing chicken. For example, authors should cite to:
Singh, M., Lim, A. J., Muir, W. I., & Groves, P. J. (2021). Comparison of performance and carcass composition of a novel slow-growing crossbred broiler with fast-growing broiler for chicken meat in Australia. Poultry Science, 100(3), 100966.
Petkov E., Ignatova M., Popova T. and Ivanova S. (2020). Quality of eggs and hatching traits in two slow-growing dual-purpose chicken lines reared conventionally or on pasture. Iranian J. Appl. Anim. Sci. 10(1), 141-148.
de Jong, I. C., Blaauw, X. E., van der Eijk, J. A., da Silva, C. S., van Krimpen, M. M., Molenaar, R., & van den Brand, H. (2021). Providing environmental enrichments affects activity and performance, but not leg health in fast-and slower-growing broiler chickens. Applied Animal Behaviour Science, 241, 105375.
Response 4: We added these references in the manuscript as the reviewer’s suggestion. Please see the revised manuscript.
Point 5: A detailed "Conclusion" should be provided to state the final result that the authors have reached. Please note you only need to place your conclusion and not keep putting results, because these have already been presented in the manuscript.
Response 5: we have revised the conclusion as the reviewer’s suggestion as follows “The results of this study indicated that the different environmental effects (on-site farm and on-station) affect both the growth characteristics and genetic expression of Thai native chickens. The phenotypic and genetic growth traits of native chickens raised on-station showed better correlations than those of native chickens raised on-site farm. In addition, the genetic parameters and breeding values in on-station showed that the breeding program by selection index for improving growth performance is valid. Therefore, to implement such a breeding program in on-site farm, the intensive or semi-intensive farm system should be considered to minimize the effect of genotype-environment interaction.” Please see the conclusion section.
Point 6: Author(s) should re-format the references based on journal format. See the instructions for authors.
Response 6: the reference has been reformatted to the journal format as the reviewer’s suggestion. Please see the references section.
Point 7: The numbers and decimals in Tables should be follow the rule of: xxxx, xxx, xx.x, x.xx, 0.xxx and 0.0xxx
Response 7: we have revised the numbers and decimals in Tables follow by the veterinary sciences journal. Please see Table 1.
Best regards,
Dr.Wuttigrai Boonkum
Corresponding author

Reviewer 3 Report
I believe data could be further maximized if ADG was broken down into growth phases (i.e. 0-4, 4-8, 8-12, and 12-16 weeks of age). The way the data is currently analyzed could be masking effects from each phase of growth, aside from the 0-4 week. I suggest at a minimum, the research group should rerun the analysis to see if any differences arise.
Line 16 & Line 24: Chicken body weight at “birth” might not be the best choice of words.
Line 84: Please elaborate on research facilities and on-sit farm management.
Line 93: The “farmer farms..” should be reworded. Perhaps On-site farm
Line 99: I understand two diet formulas were used but it could be inferred that there were two experimental treatments, instead of the intended two-phase (or stage) feeding program.
Line 100-103: This sentence could be clarified a little. The term “feed” or “diet” was missing from the first part of the sentence and “until the chickens were sold” sounds unscientific.
Line 224: I’m not certain how formatting will change for the final draft, but be mindful of starting a section at the bottom of a page.
Line 318: You touch on this later in the manuscript but I suggest elaborating more on bird management (i.e. lighting, ventilation, bedding, and stocking density) to explain performance differences with onsite vs farm raised chickens.
Line 321: I don’t disagree with the sentence, but someone could assume that greater attention to care would equate to better performance. Which wasn’t the case.
Author Response
Dear Reviewer,
We are grateful for the critical reading and your efforts to improve the quality of the manuscript. For the questions and comments, we have responded to each of you individually according to the attached file and as shown below. We hope that the manuscript in its revised form will please you.
Response to Reviewer 3 Comments
Point 1: I believe data could be further maximized if ADG was broken down into growth phases (i.e. 0-4, 4-8, 8-12, and 12-16 weeks of age). The way the data is currently analyzed could be masking effects from each phase of growth, aside from the 0-4 week. I suggest at a minimum, the research group should rerun the analysis to see if any differences arise.
Response 1: We conducted a new analysis of growth performance and genetic parameters by categorizing the average daily gain trait according to your recommendation (ADG0-4, ADG4-8, ADG8-12, and ADG12-16). We found that the new values were slightly increased from the previous values; while there were no changes in terms of statistical differences (p-value), and the conclusion was still the same. Please see the revised manuscript.
Point 2: Line 16 & Line 24: Chicken body weight at “birth” might not be the best choice of words.
Response 2: We have changed the word “body weight at birth” to “birth weight” wherever it appears in the manuscript. Please see lines 18 and 26, and the revised manuscript.
Point 3: Line 84: Please elaborate on research facilities and on-site farm management.
Response 3: We have elaborated on research facilities and on-site farm management as the reviewer’s suggestion Please see lines 119-128.
Point 4: Line 93: The “farmer farms.” should be reworded. Perhaps On-site farm.
Response 4: We have changed the word “on-farm” to “on-site farm throughout the manuscript. Please see the revised manuscript.
Point 5: Line 99: I understand two diet formulas were used but it could be inferred that there were two experimental treatments, instead of the intended two-phase (or stage) feeding program.
Response 5: We apologize for the unclear of our writing. We have revised on animal feeding more clearly. Please see lines 130-136.
Point 6: Line 100-103: This sentence could be clarified a little. The term “feed” or “diet” was missing from the first part of the sentence and “until the chickens were sold” sounds unscientific.
Response 6: The subtitle of Animal feeding was created in the revised MS. See line 126. Also, the “until the chickens were sold” was changed to “was used until the end of the experiment”. Please see lines 132-136.
Point 7: Line 224: I’m not certain how formatting will change for the final draft, but be mindful of starting a section at the bottom of a page.
Response 7: Thank you for the formatting advice. We are going to check the final draft carefully.
Point 8: Line 318: You touch on this later in the manuscript but I suggest elaborating more on bird management (i.e. lighting, ventilation, bedding, and stocking density) to explain performance differences with onsite vs farm raised chickens.
Response 8: we have revised the sentences and have elaborated more on bird management (i.e. lighting, ventilation, bedding, and stocking density) to explain performance differences between onsite vs farm raised chickens as follows:
“The low body weight in native chickens raised on-site farms may be due to different management factors, such as feed shortages in quantity and insufficient nutrients supplied for growth, and the high prevalence of diseases and parasites that usually prevail under such a system [31-33]. In addition, the differences between on-station and on-site farm management can be observed and contributed to the differences in growth characteristics as follows: the lighting, ventilation system, chicken stocking density, and bedding materials are management factors that were found to be involved in differences in growth characteristics on both farms, particularly those on-site farms were found insufficient or absent at the farm. Research indicated that light (in terms of the light source and period) was a crucial environmental factor affecting poultry performance, immune response, livability, and health status [34,35]. At the same time, the air velocity by natural ventilation may not be sufficient to reduce the air pollution of the chicken house, such as ammonia, dust, and other gas, which can influence production performances and the chicken's health [36]. Therefore, using exhaust fans, especially in hot weather, can help chickens be healthy and have better productivity [37]. Meanwhile, several studies have been conducted to study the effect of stocking density on chicken production and performance and showed the benefits of reducing stocking density on the performance of broilers [38,39]. Abudabos et al. [40] concluded that increasing the stocking density rate from 28 to 40 kg of BW/m2 had evident impingement effects on broiler chicken performance and could jeopardize their welfare. For bedding, it was found that there was not much difference between sawdust and rice husk [41]; however, choosing a material readily available in the local area was probably the best option in terms of handling.” Please see lines 399-420.
Point 9: Line 321: I don’t disagree with the sentence, but someone could assume that greater attention to care would equate to better performance. Which wasn’t the case.
Response 9: we have revised those sentences to “One of the things that support differences in growth characteristics in both farms is that when we finished the experiment, we also interviewed about the satisfaction of the Pradu Hang dum chicken farmers in the experimental area (on-site farms). We received interesting information from farmers that some farmers themselves had other activities besides raising chickens, such as rice planting and cassava farming. Therefore, it may be one reason farmers do not fully spend time raising native chickens [42,43].” Please see lines 421-426.
Best regards,
Dr.Wuttigrai Boonkum
Corresponding author
